# Association of Serum Vaspin Concentration with Metabolic Disorders in Obese Individuals

**DOI:** 10.3390/biom13030508

**Published:** 2023-03-10

**Authors:** Łukasz Pilarski, Marta Pelczyńska, Anna Koperska, Agnieszka Seraszek-Jaros, Monika Szulińska, Paweł Bogdański

**Affiliations:** 1Chair and Department of Treatment of Obesity, Metabolic Disorders and Clinical Dietetics, Poznan University of Medical Sciences, Szamarzewskiego 84 Street, 60-569 Poznań, Poland; 2Department of Bioinformatics and Computational Biology, Poznan University of Medical Sciences, Bukowska 70 Street, 60-812 Poznań, Poland

**Keywords:** adipose tissue, adipokines, vaspin, obesity, insulin resistance, anthropometry

## Abstract

Vaspin, a molecule produced in visceral adipose tissue, seems to participate in the pathogenesis of metabolic disorders. The study aimed to determine the association of vaspin concentration with metabolic disorders in obese individuals. Forty obese patients and twenty normal-weight subjects underwent biochemical (fasting glucose, insulin, lipid profile, interleukin-6, hs-CRP, vaspin concentration), blood pressure, and anthropometric measurements. The HOMA-IR index was calculated. Serum vaspin concentrations in the obese group were significantly higher than in the control group (0.82 ± 0.62 vs. 0.43 ± 0.59; *p* < 0.001). Among the entire population, vaspin concentration was positively correlated with body weight, BMI, WHR, and the percentage and mass of adipose tissue. Positive correlations between vaspin concentration and triglyceride level, insulin concentration, and HOMA-IR value were found. Vaspin concentration was positively correlated with hs-CRP and IL-6 levels. In obese patients, positive correlations between vaspin concentration and the percentage of adipose tissue and hs-CRP level were demonstrated. Logistic regression analysis showed that increased BMI was the biggest factor stimulating vaspin concentrations (OR = 8.5; 95% CI: 1.18–61.35; *p* = 0.0338). An elevated vaspin level may imply its compensatory role against metabolic disorders in obese patients. Thus, vaspin appears to be a useful diagnostic parameter for new therapeutic approaches in obesity-related complications. Nevertheless, due to the small sample size, further studies are needed to confirm our results.

## 1. Introduction

Epidemiological studies have demonstrated that obesity is a constantly growing problem that has reached the status of a global epidemic. According to the World Health Organization (WHO), the number of obese people has tripled in the last 20 years. Nowadays, obesity is defined and characterized as a disease [1].

There are a number of causes leading to the development of obesity, including genetic and environmental factors. However, the main reasons for the increasing number of people with obesity are mainly poor eating habits and a sedentary lifestyle [2]. Obesity is a state of an excessive fat accumulation caused by a disruption of energy balance due to positive caloric intake, which boosts the risk of numerous diseases [3]. While genetic factors may be involved in the pathogenesis of obesity, environmental factors contribute mainly to the presence of obesity-related disorders [4]. Obesity is a main health risk factor. It is strongly associated with the development of different pathologies, such as insulin resistance (IR), which in turn play a fundamental role in the pathogenesis of obesity-related complications, including metabolic syndrome (MetS) components, i.e., type 2 diabetes (T2DM) and dyslipidemia. Moreover, obesity may escalate cardiovascular disease (CVD), low-grade inflammation, non-alcohol fatty liver disease (NAFLD), and some types of cancer (e.g., colorectal cancer) [4]. Finally, obesity-related cardiometabolic complications are associated with increased mortality [5].

Adipose tissue is a highly active endocrine organ that regulates energy homeostasis through the secretion of numerous bioactive molecules (adipokines) [6]. Altered adipose tissue function participates in metabolic disorders once fat accumulation has begun. An excess of adipose tissue causes adipocyte synthesis dysfunction, macrophage infiltration, and low-grade inflammation. Disorder in adipokine production accompanied by obesity leads to changes both in carbohydrate and lipid metabolism [1,2]. Although the link between obesity and disturbances in adipokine production have been recognized, further studies are needed to fully understand their mechanism of action and therapeutic value [4].

Visceral adipose tissue-derived Serpin (vaspin) was first described by Hida et al. as an insulin-sensitizing adipokine secreted from the white adipose tissues (WAT) of Otsuka Long–Evans Tokushima Fatty (OLETF) rats, an animal model for obesity and T2DM [7]. It was reported that both tissue expression and serum levels of vaspin paralleled the degree of obesity and IR. Administration of vaspin to obese mice fed a high-fat/high-sucrose diet improved both insulin sensitivity and glucose tolerance and normalized blood glucose. Vaspin administration resulted in the suppression of proinflammatory adipokines, such as tumor necrosis factor α (TNF-α), resistin, and leptin, while upregulating levels of adiponectin and glucose transporter type 4 (GLUT4) in the WAT of obese mice fed a high-fat/high-sucrose diet [7]. In humans, serum concentrations of vaspin range from 0.2 to 2.5 ng/mL [8]. Elevated serum vaspin levels in humans are correlated with body mass index (BMI) and IR, and low serum vaspin concentrations represent a risk factor for the progression of T2DM. A number of studies have confirmed higher serum vaspin concentrations in obese and T2DM patients [6,9,10].

The expression of vaspin is tissue-specific and the highest level is observed in WAT [6]. Physical activity, dietary modification, and medical therapy of obese or diabetes patients affect vaspin secretion [6]. A body of evidence has confirmed that serum vaspin concentrations decrease with weight reduction and lifestyle modifications in obese adults [8]. Due to its insulin-sensitizing functions during the hyperglycemic state and protective role in vasculature and adipose states, vaspin has been speculated as a beneficial adipokine and therapeutic candidate in metabolic disorders, thus providing the reasoning for undertaking this study [4].

This study aimed to determine the association of serum vaspin concentration with metabolic disorders in obese individuals.

## 2. Materials and Methods

### 2.1. Study Population

The recruitment process was performed in the Hypertension and Metabolic Disorders Outpatient Clinic, Clinical Hospital of the Transfiguration of the Lord, Poznan, Poland. Of the recruited 60 participants, 40 were assigned to the study group of obese patients and 20 were assigned to the control group of normal-weight subjects. Both groups were comparable concerning age and sex. Participation in the study was voluntary. All participants received extensive information about the study and signed a written consent form. The study was approved by the Poznan University of Medical Science’s Bioethics Commission (approval no. 324/14). The study was conducted in accordance with the Helsinki Declaration.

The inclusion criteria for the study group were as follows: obesity defined by BMI equal to or greater than 30 kg/m^2^, age between 18–65 years, stable body mass during the last 4 weeks (±3 kg), and written consent to participate in the study. The control group consisted of healthy, normal-weight subjects with similar age and body mass stability restrictions.

The exclusion criteria included: the presence of secondary obesity, type 2 diabetes, insufficiently controlled hypertension, clinically overt atherosclerotic disease, chronic kidney disease, clinically relevant liver dysfunction, acute or chronic clinically overt inflammation process, diagnosed neoplastic disease, alcohol abuse, or cigarette smoking.

### 2.2. Study Design

#### 2.2.1. Anthropometric Parameters

Anthropometric measurements, including waist and hip circumference, were taken using a standard medical tape measure. Waist circumference was measured midway between the costal arch and upper iliac crest, and hip circumference was measured at the level of the greater trochanters. Body weight and height were measured using a RADWAG WPT 100/200 OW electric scale. The measurements were performed in the morning after overnight fasting with patients dressed only in light clothes. Height, waist circumference, and hip circumference were all determined with an accuracy of 0.5 cm. BMI and WHR (waist-hip ratio) were calculated from the received data using the appropriate formulas, and body composition (i.e., body adipose tissue, lean body mass) was assessed using the bioimpedance method with Bioscan 920–2 device (Maltron International, Rayleigh, UK). Patients were advised to avoid consuming large amounts of fluid before the test and to discontinue intense physical exercise 12 h before measurement.

#### 2.2.2. Blood Pressure Measurement

Blood pressure measurements were taken 3 times in a seated position at 2 min intervals using an ESH-validated electronic sphygmomanometer (705IT, Omron Corporation, Kyoto, Japan), according to current European Society of Hypertension (ESH) and European Society of Cardiology (ESC) [11] recommendations. Average values of systolic (SBP) and diastolic (DBP) blood pressure were calculated from the three measurements.

#### 2.2.3. Biochemical Parameters

A 5 mL sample of blood was taken from every patient on an empty stomach (defined as 12 h after last meal), which was then centrifuged and frozen at a temperature of −80 °C. Serum vaspin concentration was assessed using the human visceral adipose-specific serine protease inhibitor (vaspin) ELISA Kit (QY-E02112; Qayee-Bio, Shanghai, China) according to the manufacturer’s guidelines. The plate coefficient of variation was less than 15%. The measurement of vaspin concentration was performed in duplicate and the mean concentration was given. Serum interleukin-6 (IL-6) concentration was measured using the human interleukin-6 (IL-6) ELISA Kit (QY-E04262; Qayee-Bio, Shanghai, China). The plate coefficient of variation was also less than 15%. The concentration of IL-6 was also measured twice and the mean concentration was taking into account. High sensitivity CRP was assessed using the human high-sensitivity C-reactive protein (hs-CRP) ELISA Kit (Shanghai Sunred Biological Technology Co., Ltd., Shanghai, China). The sensitivity of the hs-CRP test was 0.112 mg/L. Sample linear regression with the expected concentration gave a correlation coefficient R over 0.95. Other biochemical analyses were performed using standard commercial tests, including: fasting glucose, insulin, and lipid profile. HOMA-IR was calculated using the appropriate formula.

#### 2.2.4. Statistical Analysis

Statistical analyses were performed using the Statistica data analysis software system version 13, 2017 (TIBCO Software Inc., Tulsa, OK, USA). The results are presented as the average ± standard deviation (SD). Compatibility of the parameters’ distribution with normal distribution was checked using the Shapiro-Wilk test. Parameters compatible with normal distribution were compared using the Student’s *t*-test for independent samples. Parameters not compatible with normal distribution were compared using the Mann-Whitney U test. Correlations between variables obeying normal distribution were assessed using the Pearson’s linear correlation coefficient. Spearman’s rank correlation coefficient was used for the remaining variables. Correlations were adjusted for age and for age and BMI. Logistic regression analysis was used to investigate if vaspin concentration was associated with elevated or lower risk of obesity. Logistic regression was performed using MedCalc^®^ Statistical Software version 20.027 (MedCalc Software Ltd., Ostend, Belgium). All differences were considered statistically significant at a level of *p* < 0.05. The Bonferroni-Hochberg correction was applied.

The sample size was determined according to the vaspin concentration, based on the study by Tarabeih et al. [12]. The mean vaspin vespin concentrations in the study (LBP-Duration) and control groups were 6.11 ± 0.074 and 5.83 ± 0.044 pg/mL, respectively. It was calculated that a sample size of at least 10 subjects in each group would yield at least 80% power in detecting a significant difference.

## 3. Results

A total of 40 people aged 43.3 ± 13.4 years were enrolled into the study group while 20 healthy participants aged 38.9 ± 14.7 years were recruited to the control group. The patients from the obese group had significantly higher body weights (by 35.9%, *p* < 0.001), BMI values (by 44.1%, *p* < 0.001), WHR (by 17.2%, *p* = 0.001), and percentages of body adipose tissue (43.3%, *p* < 0.001) than the subjects from the control group (Table 1). Among the biochemical parameters, differences occurred in the concentration of triglycerides (by 113.3%, *p* < 0.001), hs-CRP (68.7%, *p* < 0.001), and IL-6 (53.4%, *p* < 0.001), which were higher in the obese group. A similar situation occurred with IR markers, with higher insulin levels (52.9%, *p* = 0.004) and HOMA-IR values (78.0%, *p* < 0.001). In contrast, levels of HDL cholesterol were significantly lower in the study group than in the control group (by 32.3%, *p* < 0.001). There were no differences in reference to total cholesterol and LDL levels, glucose concentration, and blood pressure values between groups. The serum vaspin concentrations in the obese group were significantly higher than in the control group (0.82 ± 0.62 vs. 0.43 ± 0.59; *p* < 0.001). Females and males did not differ in vaspin concentration. The detailed characteristics of the study population are shown in Table 1.

In the entire population, vaspin concentration was positively correlated with body weight (r = 0.452; *p* = 0.003), BMI (r = 0.558; *p* < 0.001, Figure 1), WHR (r = 0.447; *p* = 0.003, Table 2), and the content of body adipose tissue (both percentage and mass; r = 0.616 and r = 0.507, respectively; *p* < 0.001, Figure 2). For the lipid metabolism parameters, a statistically significant positive correlation was found between vaspin concentration and triglyceride level (r = 0.337; *p* = 0.013). In contrast, no significant associations were found between vaspin concentration and total cholesterol, HDL, or LDL concentrations. We found a positive correlation between vaspin concentration and IR markers such as insulin level (r = 0.341; *p* = 0.013) as well as HOMA-IR value (r = 0.382; *p* = 0.022), with no associations with fasting glucose level. Moreover, vaspin concentration was positively correlated with hs-CRP (r = 0.614; *p* < 0.001) and IL-6 levels (r = 0.457; *p* = 0.003, Table 2). After adjusting for age, vaspin concentration was still correlated with BMI and WHR values, the amount of body adipose tissue, as well as insulin and hs-CRP concentrations. Meanwhile, after adjusting for age and BMI, vaspin concentration was only correlated with hs-CRP level (Table 2).

In obese patients, positive correlations between vaspin concentration and the percentage of body adipose tissue (r = 0.382; *p* = 0.030) and hs-CRP level (r = 0.428; *p* = 0.002, Table 3) were demonstrated. No statistically significant relationships were found with the control group.

The results of logistic regression showed that a one unit increase in vaspin concentration increased the risk of obesity (based on BMI) 8.5 times (OR = 8.5; 95% CI: 1.18–61.35; *p* = 0.0338; Table 4). The statistical significance of this association was maintained after adjusting the model for age (OR = 8.33; 95% CI: 1.15–60.21; *p* = 0.0338).

## 4. Discussion

The present study documented higher vaspin concentrations in obese individuals than in normal-weight subjects. This study also showed that BMI value was an independent determinant of serum vaspin concentration. Moreover, several positive correlations between serum vaspin concentration and cardiometabolic risk-related parameters were observed in obese patients and the entire studied population.

The research data confirmed the relationship between vaspin concentration and obesity. Similarly to our study, Taheri et al. demonstrated higher serum vaspin levels in normal-weight obese patients than in non-obese controls [13]. Yang et al., in a study involving 66 patients with type 2 diabetes (including 36 obese subjects) and 48 patients without diabetes (including 21 obese subjects), found higher vaspin concentrations in obese patients, both in the group with diabetes and that with normal carbohydrate metabolism [14]. Additionally, a meta-analysis of six studies evaluated the significantly higher serum vaspin levels in obese (*n* = 1826, vaspin level higher by 0.52 ng/mL, 95% confidence interval [CI]: 0.10–0.93, *p* = 0.02) and T2DM patients (*n* = 1570, vaspin level higher by 0.36 ng/mL, 95% CI: 0.23–0.49, *p* < 0.00001) that in non-obese healthy controls [6]. On the other hand, not all analyses have confirmed these reports. In a study conducted by Auguet et al. involving 71 women, including 40 morbidly obese (BMI > 40 kg/m^2^) and 31 normal body-weight (BMI < 25 kg/m^2^) patients, no significant difference in serum vaspin concentration was found. However, the authors reported significantly higher mRNA expression of this adipokine in visceral (VAT) and subcutaneous adipose tissue (SAT) in women with obesity [15]. It is worth mentioning that in contrast to our study, previous research [9,16,17] has shown sexual dysmorphism in relation to vaspin concentrations. Some authors attributed the increased vaspin level in females to high estrogen concentrations [18]. However, several studies found no gender differences regarding vaspin concentration [19,20,21]. It has been also indicated that sex differences are abrogated in type 2 diabetic patients [9]. Thus, the absence of sexual dimorphism regarding the discussed molecule may be associated with metabolic disturbances, such as impaired glucose tolerance or insulin sensitivity, which may explain the lack of gender difference in our study.

As previously mentioned, vaspin is expressed mainly in adipose tissue (visceral and subcutaneous) in humans, although its expression is also noted in many other tissues, such as the liver, pancreas, skin, placenta, stomach, cerebrospinal fluid, hypothalamus, and ovaries [22]. Its level depends on different genetic and environmental factors, among which the most important seem to be body weight and the content of adipose tissue. Both vaspin mRNA and serum levels are associated with obesity and impaired insulin sensitivity. The level of vespin mRNA in adipose tissue is correlated with body mass increase [23] and is elevated in patients with obesity and T2DM [9,10]. Other factors that increase the expression of this adipokine in VAT and serum are insulin and leptin levels, the presence of IR, as well as exposure to a high-fat diet [24]. Gonzales et al. documented that age, gender, and nutritional status may also affect vaspin expression in WAT [25].

It is documented that vaspin may have positive effects on glucose and insulin metabolism, lipid profile, appetite control, and arteriosclerosis, thus counteracting obesity, IR, and inflammation [26]. In the already mentioned study conducted by Hida et al., it was shown that vaspin was poorly detectable in young OLETF rats and concentrations increased with age, weight, and insulin level, peaking at 30 weeks, when rats achieved the highest body weight and IR. On the other hand, the authors observed a reduction in vaspin concentration together with aggravation of T2DM and body weight loss in OLETF rats [7]. Other studies reported that central and peripheral vaspin administration led to a reduction in food intake [27] and increased insulin sensitivity in both db/db and C57BL6 mice [28]. Conversely, vaspin knockout mice consuming a high-fat/high-sucrose diet showed increased body weight, macrophage infiltration in adipose tissue, lipid accumulation in the liver, and aggravation of insulin sensitivity [29]. Thus, an elevated vaspin level in subjects with excessive body weight may imply the compensatory role of vaspin in obesity and metabolic dysfunction. The potential anorexigenic effects of vaspin may result from inhibition of a protease degrading a putative anti-orexigenic factor [30]. Moreover, it was reported that administration of vaspin into the arcuate nucleus (ARC) of the rat hypothalamus reduced hunger and food intake through the significant decrease in neuropeptide Y (NPY) and increase in proopiomelanocortin gene expression [31]. In turn, vaspin-induced antidiabetic effects include the promotion of islet cell secretion in the pancreas, protection of β cells from damage mediated by nuclear factor-kappa B (NF-κB), as well as reduction of hepatic glucose production [32].

In our entire study population, vaspin concentration was positively correlated with cardiometabolic risk-related parameters, such as body weight, BMI, and percentage of adipose tissue. Those correlations (besides body weight) were still significant after adjusting for age. On the other hand, due to intercorrelations between the obesity-related variables, after adjusting for age and BMI, the correlations between anthropometric and biochemical parameters and vaspin level were not statistically significant. Similarly to our results, Yang et al. showed positive relationships between serum vaspin concentration and BMI, fat percentage, and triglyceride level [14]. In another study, strong positive correlations between vaspin concentration and BMI and waist circumference were demonstrated in patients with sleep apnea [33]. On the other hand, Alizadeh et al. did not confirm this relationship, showing no correlations between serum vaspin level and body composition components (BMI, fat percentage) [34]. It is worthwhile to note that BMI increase was the biggest factor stimulating vaspin concentrations in obese individuals in the current study, showing that anthropometric measurements are useful tools in the diagnosis of excessive body weight.

The present study showed positive correlations between vaspin concentrations and IR marker indicators such as fasting insulin and HOMA-IR in the entire study population. Studies on the relationship between vaspin level and carbohydrate metabolism parameters provide various information. In line with our study, Suleymanoglu et al. showed that vaspin level was positively correlated with fasting insulin level and HOMA-IR value [20]. Additionally, other authors demonstrated significant associations between vaspin concentration and fasting plasma insulin level as well as the insulin sensitivity index. Moreover, it has been indicated that the latter is an independent factor affecting the level of the discussed adipokine [14]. The positive correlations between vaspin level and insulin concentration, HOMA-IR index, and fasting glucose level were also confirmed in the study by Aliasghari et al. on a group of 83 patients with nonalcoholic fatty liver disease (NAFLD) [35]. On the contrary, not all studies have confirmed these results [17,36]. Interesting results were provided in a study by Wada et al. These authors showed a decrease in serum serum vaspin concentrations in OLETF rats together with the deterioration of diabetes control. The use of pioglitazone caused an increase in the concentration of this adipokine, and the administration of recombinant vaspin to OLETF rats improved their glucose tolerance and insulin sensitivity [37]. The exact effect of vaspin’s action on insulin signalling and metabolism in humans is not fully understood. It is indicated that vaspin may influence glycemic control by kallikrein 7, a human protease that cleaves insulin in vitro. Vaspin, through its serpin activity, seems to inhibit kallikrein 7, thus the antidiabetic effect of this adipokine may be related to decreased degradation of circulating insulin [28]. Moreover, vaspin may mediate insulin signalling via binding to glucose-related protein (GRP78), 7 KDa voltage-dependent anion channel. It was demonstrated that vaspin binds GRP78 in HepG2 cells. Additionally, the stimulation of H−4-II-E-C3 cells with recombinant vaspin led to activation of the protein kinase B (AKT) and the 5’AMP-activated protein kinase (AMPK) signalling pathways, which were disturbed by inhibition of GRP78 [29]. Recently, GRP78 has been shown to be highly expressed in adipose tissue both in humans and mice and levels increase with age, obesity, and diabetes. The overexpression of GRP78 seems to be related to hyperinsulinemia in adipocytes, thus the management of GRP78 expression may be a potential therapeutic target against disturbed carbohydrate metabolism [38].

Our study indicated a positive correlation between serum vaspin concentration and hs-CRP level both in the entire population and the study group. This correlation was still significant even after adjustment for age as well as age and BMI. Moreover, a positive correlation between vaspin concentration and IL-6 level was demonstrated in the whole study population. A significant positive relationship between the concentration of the discussed adipokine and hs-CRP level was also documented by other authors in children with metabolic syndrome [10] and women with polycystic ovary syndrome [39]. An anti-inflammatory effect of vaspin has been suggested, probably due to its ability to inhibit the expression of proinflammatory adhesion markers, such as intercellular adhesion molecule 1 (ICAM1), vascular cell adhesion molecule 1 (VCAM1), E-selectin, and monocyte chemoattractant protein 1 (MCP-1) [40]. Vaspin attenuates the expression of adhesion molecules in the AMPK-dependent pathway and inhibits the activity of NF-κB [41,42]. Vaspin reduces the formation of reactive oxygen species (ROS) and apoptosis induced by oxidative stress of mesenchymal stem cells (MSCs) [43,44]. Vaspin also increases the bioavailability of nitric oxide (NO) by increasing the expression of DDAH II (dimethylarginine dimethylaminohydrolase)–the enzyme responsible for the degradation of ADMA (asymmetric dimethylarginine), which has an inhibitory effect on endothelial NO synthase [45]. Furthermore, vaspin seems to modulate lipid metabolism. Its infusion decreased triglyceride and free fatty acid levels and promoted cholesterol efflux in macrophages due to the upregulation of ATP-binding cassette transporter A1 (ABCA1) [46,47]. The relationship between vaspin and lipid parameters was also documented in our study.

It is important to note that our study has some limitations. The first limitation is the size of the study and control groups. To ensure a homogeneous study population, we applied numerous exclusion criteria. Moreover, the study included only Caucasians; therefore, the generalization of the results should be done with caution. Secondly, to assess body composition we used electrical bioimpedance together with anthropometric measurements instead of magnetic resonance imaging or the DEXA method. Nevertheless, all of the mentioned methods are widely accepted in clinical practice. Further, the assessment of IR was based on the value of the HOMA-IR index not the euglycemic clamp method due to its invasiveness and the duration of the measurement. In addition, this was an observational study and was thus unable to identify genetic variations between vaspin concentration and the analyzed metabolic parameters. Therefore, due to small sample size, the provided results should be considered as preliminary and further studies with a bigger sample size are needed to confirm our report as well as elucidate the molecular mechanisms underlying the relationship between vaspin level and cardiometabolic risk-related parameters.

On the other hand, the key strengths of the study include a deep analysis of biochemical, physiologic, and anthropometric parameters and their association with metabolic disorders in obese individuals. Thus, our study provides insight into the role of vaspin in the pathogenesis of obesity-related complications.

## 5. Conclusions

Obese individuals presented higher serum vaspin levels than normal-weight subjects. Several positive correlations between vaspin level and cardiometabolic risk-related parameters were found. Furthermore, BMI value turned out to be an independent determinant of serum vaspin concentration in obese individuals. Thus, an elevated vaspin level in subjects with excessive body weight may imply the compensatory role of the discussed adipokine against obesity and its complications. In conclusion, vaspin appears to be a useful diagnostic parameter for new therapeutic approaches in obesity-related disorders. Nevertheless, further research with a larger cohort is needed to confirm the results obtained in our study.

## Figures and Tables

**Figure 1 biomolecules-13-00508-f001:**
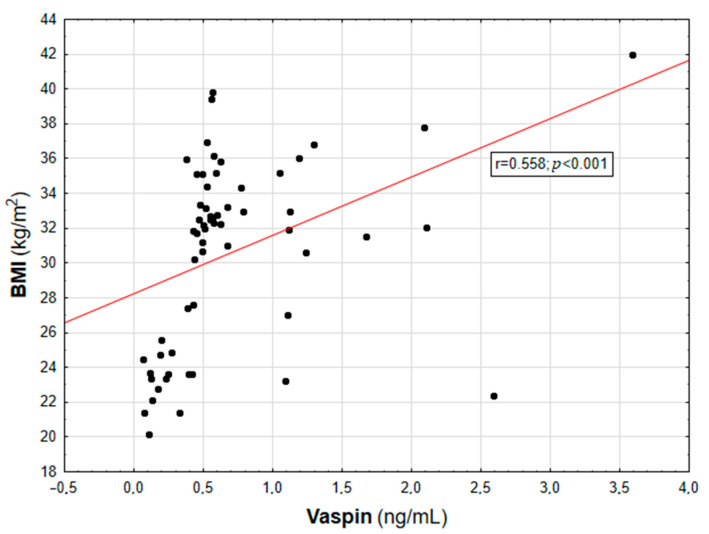
Correlation graph showing the relationship between serum vaspin concentration and BMI value.

**Figure 2 biomolecules-13-00508-f002:**
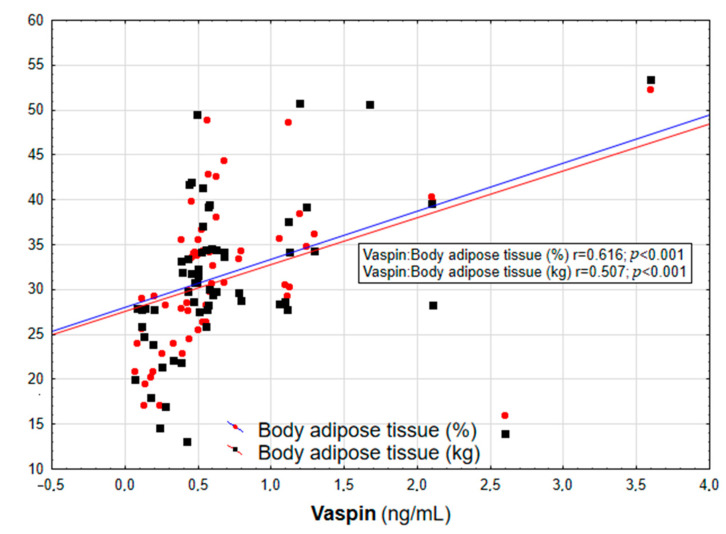
Correlation graph showing the relationship between serum vaspin concentration and body adipose tissue content (in % and kg).

**Table 1 biomolecules-13-00508-t001:** Clinical characteristics of the study population.

Parameter (Unit)	Obese Group (*n* = 40)		Control Group (*n* = 20)		*p*-Value
	Mean	SD	Mean	SD	*p*
Female/Male	20/20	-	10/10	-	NS
Age (years)	43.3	13.4	38.9	14.7	NS *
Body weight (kg)	95.3	8.8	70.13	8.65	<0.001 #
BMI (kg/m^2^)	33.9	2.7	23.53	1.9	<0.001 *
WHR	1.02	0.14	0.87	0.14	0.001 *
Adipose tissue (%)	35.4	6.3	24.7	5.7	<0.001 #
Adipose tissue (kg)	35.1	6.9	23.4	5.7	<0.001 *
SBP (mmHg)	132	12	132	6	NS *
DBP (mmHg)	79	7	78	7	NS #
Total cholesterol (mmol/L)	5.59	1.11	5.12	0.38	NS #
LDL (mmol/L)	3.59	1.14	3.17	0.53	NS #
HDL (mmol/L)	1.05	0.40	1.55	0.32	<0.001 #
Triglycerides (mmol/L)	2.07	0.87	0.97	0.45	<0.001 *
Glucose (mmol/L)	5.18	0.64	4.93	0.53	NS *
Insulin (µU/mL)	15.19	5.99	9.94	2.95	0.004 *
HOMA-IR	3.65	1.55	2.05	0.63	<0.001 *
hs-CRP (mg/L)	2.80	0.79	1.66	0.66	<0.001 #
IL-6 (pg/mL)	4.31	0.62	2.81	0.68	<0.001 *
Vaspin (ng/mL)	0.82	0.62	0.43	0.59	<0.001 *

SD–standard deviation; *p*–statistical significance; NS–not significant; BMI–body mass index; WHR–waist-hip ratio; SBP–systolic blood pressure; DBP–diastolic blood pressure; LDL–low-density lipoprotein; HDL–high-density lipoprotein; TG–triglycerides; HOMA–IR–Homeostatic Model Assessment of Insulin Resistance; hs-CRP–high-sensitivity C-reactive protein; IL-6–interleukin-6; * Mann-Whitney U test; # Student’s *t*-test.

**Table 2 biomolecules-13-00508-t002:** Correlations between vaspin concentration and estimated parameters in the whole population.

Variables (Unit)	Correlation Coefficient (Rho)	*p*-Value *	Correlation Coefficient (Rho) Covariates-Age	*p*-Value *	Correlation Coefficient (Rho) Covariates–Age, BMI	*p*-Value *
Body weight (kg)	0.452	0.003	0.198	NS	−0.153	NS
BMI (kg/m^2^)	0.558	<0.001	0.393	0.004	-	-
WHR	0.447	0.003	0.398	0.004	0.247	NS
Body adipose tissue (%)	0.616	<0.001	0.431	<0.001	0.225	NS
Body adipose tissue (kg)	0.507	<0.001	0.385	0.017	0.175	NS
Total cholesterol (mmol/L)	0.211	NS	0.105	NS	0.033	NS
LDL cholesterol (mmol/L)	0.184	NS	0.119	NS	0.065	NS
HDL cholesterol (mmol/L)	−0.243	NS	−0.134	NS	0.019	NS
Triglycerides (mmol/L)	0.337	0.013	0.089	NS	−0.099	NS
Glucose (mmol/L)	0.141	NS	0.175	NS	0.084	NS
Insulin (µU/mL)	0.341	0.013	0.395	0.004	0.251	NS
HOMA-IR	0.382	0.022	0.374	NS	0.184	NS
hs-CRP (mg/L)	0.614	<0.001	0.554	<0.001	0.429	<0.001
IL-6 (pg/mL)	0.457	0.003	0.229	NS	−0.074	NS
SBP (mmHg)	0.001	NS	0.086	NS	0.037	NS
DBP (mmHg)	0.013	NS	0.086	NS	0.031	NS

R–correlation coefficient; R (age)–correlation coefficient after adjustment for age; R (age, BMI)–correlation coefficient after adjustment for age and BMI; *p*–statistical significance; NS–not statistically significant; BMI–body mass index; WHR–waist-hip ratio; LDL–low-density lipoprotein; HDL– high-density lipoprotein; HOMA–Homeostatic Model Assessment of Insulin Resistance; hs-CRP–high-sensitivity C-reactive protein; IL-6–interleukin-6; SBP–systolic blood pressure; DBP–diastolic blood pressure; * Spearman rank correlation test.

**Table 3 biomolecules-13-00508-t003:** Correlations between vaspin concentration and estimated parameters in the study group.

Variables (Unit)	Correlation Coefficient (Rho)	*p*-Value *
Body adipose tissue (%)	0.382	0.030
hs-CRP (mg/L)	0.428	0.002

*p*–statistical significance; hs-CRP–high-sensitivity C-reactive protein; * Spearman rank correlation test.

**Table 4 biomolecules-13-00508-t004:** Logistic regression investigating the association of vaspin concentration with obesity.

	OR	95% CI	*p*
raw	8.5	1.18–61.35	0.0338
adjusted *	8.33	1.15–60.21	0.0338

OR–Odds Ratio; CI–Confidence Interval; * adjusted for age.

## Data Availability

The data presented in this study are available upon request from the corresponding author. The data are not publicly available due to privacy restrictions.

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
