# Peer review of "Association of Serum Vaspin Concentration with Metabolic Disorders in Obese Individuals"

_biomolecules, 2023, doi:10.3390/biom13030508_

Round 1

Reviewer 1 Report

Pilarski Łukasz and colleagues measured biochemical, blood pressure, etc, in 40 obese patients and 20 normal wight subjects. They found Vaspin was positively correlated with body weight, BMI, waist circumference, the percent and mass of adipose tissue. A positive correlation between vaspin and triglycerides levels, insulin concentration and HOMA-IR value was found. Vaspin corelated positively with hs-CRP and IL-6 levels. Although the authors presented data relevantly support their conclusion, there are a couple of concerns that needs to be addressed.

In 2008, a study analyzed the serum Vaspin concentrations in human obesity and type 2 diabetes (Diabetes. 2008, 57, 372-377. doi: 10.2337/db07-429 1045.). They found circulating vaspin significantly correlated with BMI and insulin sensitivity. What’s the main difference between two studies? 

Authors found Vaspin concentration didn’t show gender difference. However, in previous study (Diabetes. 2008, 57, 372-377. doi: 10.2337/db07-429 1045.), they reported that Vaspin concentration is significantly higher in female compared with male subjects. Should the authors discuss the inconsistence? 

To better represent the data and support authors’ conclusion, scatter plot is highly recommended to organize the data. 

Author Response

Dear Reviewer,

Thank you for careful reading of our manuscript. All of the suggestion have been included in the modified version of the article and marked in red in the text. We would like to also admit, that considering all Reviewers suggestions, our manuscript was improved in general.

  • Comment no 1.

In 2008, a study analyzed the serum Vaspin concentrations in human obesity and type 2 diabetes (Diabetes. 2008, 57, 372-377. doi: 10.2337/db07-429 1045.). They found circulating vaspin significantly correlated with BMI and insulin sensitivity. What’s the main difference between two studies?

Thank you for your comment. Article form 2008 written by Youn et al, contrary to the our study, did not show the differences in circulating vaspin between individuals with normal glucose tolerance and body weight and patients with type 2 diabetes and obesity. On the other hand, authors indicated that elevated vaspin serum concentration is associated with obesity and impaired insulin sensitivity, what confirmed the results obtained in our study. We mentioned about discussed article several times in our manuscript.  

  • Comment no 2.

Authors found Vaspin concentration didn’t show gender difference. However, in previous study (Diabetes. 2008, 57, 372-377. doi: 10.2337/db07-429 1045.), they reported that Vaspin concentration is significantly higher in female compared with male subjects. Should the authors discuss the inconsistence?

The mentioned article was written by other research team (Youn et al.). Indeed, on the contrary to our results, authors found a sexual dysmorphism in relation to vaspin concentrations. Although those authors also showed that sex differences were abrogated in type 2 diabetic patients. Also other studies confirmed this findings. For example Chang et al. (Obesity (Silver Spring). 2010;18(11):2105-10. doi: 10.1038/oby.2010.60.) found no differences in vaspin concentration between male and female. Authors indicated that the absence of sexual dimorphism regarding to serum vaspin levels can be correlated with metabolic disturbances as impaired glucose tolerance or insulin sensitivity, what may explain the lack of gender difference in our study. This time, we discussed this issue in the manuscript: “It is worth mentioning, that on contrary to our study, previous research [9, 16, 17] showed a sexual dysmorphism in relation to vaspin concentrations. Some authors explained an increased vaspin level in female due to high estrogens concentration [18]. In contrast, several studies found no gender differences regarding to vaspin concentration [19, 20, 21]. It has been also indicated that the sex differences are abrogated in type 2 diabetic patients [9]. Thus, the absence of sexual dimorphism regarding to dis-cussed molecule can be associated with metabolic disturbances as impaired glucose tolerance or insulin sensitivity, what may explain the lack of gender difference in our study. (page 7 line 278).

  • Comment no 3.

To better represent the data and support authors’ conclusion, scatter plot is highly recommended to organize the data.

Thank you for your valuable comment. The graphics has been added to the result section (page 5 line 194 and page 6 line 221).

Reviewer 2 Report

Reviewer comments on "Manuscript Association of serum vaspin concentration with metabolic disorders in obese individuals" by Pilarski et al

This study compares serum vaspin concentration levels in obese individuals vs normal weight individuals. In humans, it has been repeatedly reported that visceral and subcutaneous adipose tissue express vaspin, which is presumably able to regulate insulin resistance. It has been shown that in obese laboratory animals administration of vaspin able to improve glucose tolerance and reduces food intake. In this connection, the present study is certainly of interest. The major working hypothesis of the present study is that circulating levels of vaspin are associated with metabolic disorders in obese individuals. This assumption was tested on a very modest sample, including 40 obese vs 20 normal-weight individuals. Several blood biochemical factors, such as fasting glucose and insulin, lipid profile and some cytokines, namely interleukin 6 and hsCRP were tested for association with vaspin circulating levels. Some other measurements, including anthropometric measurements were taken. The manuscript is well written and the results are clearly presented. The authors showed that despite the small sample size of the study, with just a few exceptions all the comparisons and correlations were statistically significant. This makes this study very interesting and potentially significant.

However, this reviewer has a number of concerns.

The major concern is the small sample size 40 cases, of both sexes with variety of ages and 20 control individuals.  This raises the major concern of the reliability and reproducibility of the obtained results. With such significant and consistent results as provided in Table 2, at least small replication study is required to confirm these results. This is in particular important issue because other studies, using  quite a big community-based samples found no such consistent correlations between the vaspin plasma concentrations and body anthropometrics and also body composition assessed by BIA method (e.g. https://doi.org/10.3390/diagnostics10100797). Such correlations were not observed also in case/control study that used small sample of diabetic vs control individuals (PMID: 30740159)

My other concerns and suggestions are:

1. The detailed descriptive statistics for circulating levels of vaspin, hsCRP, and IL6, including the minimal detection levels and the intra- and inter-assay coefficients of variation.

3. In the present study, "Females and males did not differ in vaspin concentration." This, however, contradicts some other studies. For example, the levels of vaspin (pg/mL) were significantly higher in females compared to males in the much bigger study (https://doi.org/10.3390/diagnostics10100797). Is this situation (no sex differences) caused by the limited power of the sample?

4. This reviewer has many years of experience in using statistical methods of analysis and is very dubious that "a sample size of at least 10 subjects in each group would yield at least 90% power of detecting the significant difference."

5. The study sample includes individuals with a wide range of ages (in the obese group mean age was 43.3 and SD -13.4). In view of this reviewer, in all the comparisons and correlations, the effect of age must be taken into account. Due to the small sample size, probably in advance adjustment for best fitting age function of each independent variable in the study would be recommended.

6. It should be mentioned that various body/fat mass measurements are expected to be highly inter-correlated (in multivariate regression analysis, they will create heavy redundancy due to collinearity) and, therefore, shouldn't be all analyzed. Probably most independent variables should be selected and examined.

7. The results of the analyses presented in Table 2 are problematic in many instances, as may be explained by their covariation with age and BMI. For example, there are many studies suggesting that hsCRP and IL6 levels correlate with BMI and other obesity indices. This would explain their and other variables' correlations with vaspin, in particular when obese and lean individuals combined. Of course, body weight and BMI, as well as waist and hip circumferences, should not be both tested, etc. WHR is probably better in any case.

Therefore, in view of this reviewer, Table 2 results presentation will benefit if two columns were added: the same type of analysis after adjustment for age and the same type of analysis after adjustment for age and BMI.

8. There are numerous tests of the same hypothesis in the study (multiple testing problem). Therefore, correction for multiple testing is required.

Author Response

Dear Reviewer,

Thank you for your careful reading of our manuscript. We agree with your concerns. The number of patients was limited due to the numerous exclusion criteria, which ensured a very homogeneous population. The study excluded people over 65 years of age due to decreases in their lean body mass, basic metabolism rate, and sensitivity of tissue to insulin action. We admit that future studies could be done with larger research and control groups to confirm the correlation between vaspin serum concentration and anthropometric as well as biochemical parameters. It will be our future direction. Nevertheless, growing body of evidences, similarly to our results suggest a strong relationship especially between vaspin level and anthropometric indicators (DOI: 10.4238/2015.September.25.2; DOI: 10.3892/etm.2016.2997).

We would like to also admit, that considering all Reviewers suggestions, our manuscript was improved in general. All of the suggestion have been included in the modified version of the article and marked in red in the text.

  • Comment no 1.

The detailed descriptive statistics for circulating levels of vaspin, hsCRP, and IL6, including the minimal detection levels and the intra- and inter-assay coefficients of variation.

The information has been added in the Materials and Methods section according to manufacturer’s guidelines (page 3 line 129). Although the producer did not give any information about the minimal detection levels for vaspin and IL-6.

  • Comment no. 3

In the present study, "Females and males did not differ in vaspin concentration." This, however, contradicts some other studies. For example, the levels of vaspin (pg/mL) were significantly higher in females compared to males in the much bigger study (https://doi.org/10.3390/diagnostics10100797). Is this situation (no sex differences) caused by the limited power of the sample?

Thank you for your valuable comment. Indeed, on the contrary to our results, some authors found a sexual dysmorphism in relation to vaspin concentrations. Although not all studies confirmed this findings. For example Chang et al. (Obesity (Silver Spring). 2010;18(11):2105-10. DOI: 10.1038/oby.2010.60.) evaluated no differences in vaspin concentration between male and female. Authors indicated that the absence of sexual dimorphism regarding to serum vaspin levels can be correlated with metabolic disturbances as impaired glucose tolerance or insulin sensitivity, what may explain the lack of gender difference in our study, independently from the power of the sample. We discussed this issue in the manuscript this time: “It is worth mentioning, that on contrary to our study, previous research [9, 16, 17] showed a sexual dysmorphism in relation to vaspin concentrations. Some authors explained an increased vaspin level in female due to high estrogens concentration [18]. In contrast, several studies found no gender differences regarding to vaspin concentration [19, 20, 21]. It has been also indicated that the sex differences are abrogated in type 2 diabetic patients [9]. Thus, the absence of sexual dimorphism regarding to dis-cussed molecule can be associated with metabolic disturbances as impaired glucose tolerance or insulin sensitivity, what may explain the lack of gender difference in our study.” (page 7 line 278).

  • Comment no. 4

This reviewer has many years of experience in using statistical methods of analysis and is very dubious that "a sample size of at least 10 subjects in each group would yield at least 90% power of detecting the significant difference."

Thank you for your comment. As we mentioned in our manuscript, the sample size was determined according to vaspin concentration based on the article described by Teshigawara et al. (DOI: 10.1210/jc.2011-3297.). The calculations showed, that a sample size of at least 10 subjects in each group would yield at least 90% power of detecting the significant difference. We verified this calculations one more time (page 4 line 153).

  • Comment no. 5

The study sample includes individuals with a wide range of ages (in the obese group mean age was 43.3 and SD -13.4). In view of this reviewer, in all the comparisons and correlations, the effect of age must be taken into account. Due to the small sample size, probably in advance adjustment for best fitting age function of each independent variable in the study would be recommended.

The correction has been added to the Table 2 (page 5 line 215).

  • Comment no. 6

It should be mentioned that various body/fat mass measurements are expected to be highly inter-correlated (in multivariate regression analysis, they will create heavy redundancy due to collinearity) and, therefore, shouldn't be all analyzed. Probably most independent variables should be selected and examined.

Thank you for your comment. We are aware that a multivariate regression may cause a heavy redundancy due to collinearity, however we use a logistic regression with one variable (vaspin concentration only).

  • Comment no. 7

The results of the analyses presented in Table 2 are problematic in many instances, as may be explained by their covariation with age and BMI. For example, there are many studies suggesting that hsCRP and IL6 levels correlate with BMI and other obesity indices. This would explain their and other variables' correlations with vaspin, in particular when obese and lean individuals combined. Of course, body weight and BMI, as well as waist and hip circumferences, should not be both tested, etc. WHR is probably better in any case.

Therefore, in view of this reviewer, Table 2 results presentation will benefit if two columns were added: the same type of analysis after adjustment for age and the same type of analysis after adjustment for age and BMI.

Table 2 has been modified as suggested. WHR has been introduced instead of waist and hip circumferences (page 5 line 215).

  • Comment no. 8

There are numerous tests of the same hypothesis in the study (multiple testing problem). Therefore, correction for multiple testing is required.

Thank you for your suggestions. Although, due to the comparative analysis between the two groups, that is the study and the control, we decided not to introduce the corrections to the testing. In the future studies, with more groups, we certainly use the correction for multiple testing.

Round 2

Reviewer 1 Report

Authors addressed all my concerns.

Author Response

We would like to thank one more time the Reviewer for the expert evaluation of our manuscript.

Reviewer 2 Report

The present version of the manuscript was substantially improved, and the authors made an effort to meet the reviewer's critical comments and suggestions. However, this reviewer believes that the revision of the manuscript is not completed and some problematic issues remained unanswered.

This reviewer added the new comments to the previous ones taking into account authors' corresponding response.

·         Comment no. 1

R: The small sample size is the major concern in this study.

AU: Thank you for your careful reading of our manuscript. We agree with your concerns. The number of patients was limited due to the numerous exclusion criteria, which ensured a very homogeneous population. The study excluded people over 65 years of age due to decreases in their lean body mass, basic metabolism rate, and sensitivity of tissue to insulin action. We admit that future studies could be done with larger research and control groups to confirm the correlation between vaspin serum concentration and anthropometric as well as biochemical parameters. It will be our future direction. Nevertheless, growing body of evidences, similarly to our results suggest a strong relationship especially between vaspin level and anthropometric indicators (DOI: 10.4238/2015.September.25.2; DOI: 10.3892/etm.2016.2997).

We would like to also admit, that considering all Reviewers suggestions, our manuscript was improved in general. All of the suggestion have been included in the modified version of the article and marked in red in the text.

R_New comment: This explanations are not very convincing, therefore some simulation or testing for minimal sample size would be required DOI: 10.3892/etm.2016.2997 is not very relevant for this study. The paper doi: 10.3390/diagnostics10100797 probably more relevant to this study.

·         Comment no. 2

R: The detailed descriptive statistics for circulating levels of vaspin, hsCRP, and IL6, including the minimal detection levels and the intra- and inter-assay coefficients of variation.

AU: The information has been added in the Materials and Methods section according to manufacturer’s guidelines (page 3 line 129). Although the producer did not give any information about the minimal detection levels for vaspin and IL-6.

R_new: The study (DOI: 10.1038/oby.2010.60) referenced by the authors states the following with respect to vaspin measurement: "The assay sensitivity was 12 pg/ml, and the intra- and interassay coefficients of variance were 1.3–3.8 and 3.3–9.1%, respectively." Also the study using much bigger sample (doi: 10.3390/diagnostics10100797) mentions: " The detection limits were 49.6 pg/mL for vaspin and 375 µg/mL for adipsin. The intra- and inter-assay coefficients of variation were between 2.3% and 8.3%". This is what was required for the present study. Note that 15% intra-assay variation seems too high .In such a case duplicate measurements strongly recommended.

·         Comment no. 3

R: In the present study, "Females and males did not differ in vaspin concentration." This, however, contradicts some other studies. For example, the levels of vaspin (pg/mL) were significantly higher in females compared to males in the much bigger study (https://doi.org/10.3390/diagnostics10100797). Is this situation (no sex differences) caused by the limited power of the sample?

AU: Thank you for your valuable comment. Indeed, on the contrary to our results, some authors found a sexual dysmorphism in relation to vaspin concentrations. Although not all studies confirmed this findings. For example Chang et al. (Obesity (Silver Spring). 2010;18(11):2105-10. DOI: 10.1038/oby.2010.60.) evaluated no differences in vaspin concentration between male and female. Authors indicated that the absence of sexual dimorphism regarding to serum vaspin levels can be correlated with metabolic disturbances as impaired glucose tolerance or insulin sensitivity, what may explain the lack of gender difference in our study, independently from the power of the sample. We discussed this issue in the manuscript this time: “It is worth mentioning, that on contrary to our study, previous research [9, 16, 17] showed a sexual dysmorphism in relation to vaspin concentrations. Some authors explained an increased vaspin level in female due to high estrogens concentration [18]. In contrast, several studies found no gender differences regarding to vaspin concentration [19, 20, 21]. It has been also indicated that the sex differences are abrogated in type 2 diabetic patients [9]. Thus, the absence of sexual dimorphism regarding to discussed molecule can be associated with metabolic disturbances as impaired glucose tolerance or insulin sensitivity, what may explain the lack of gender difference in our study.” (page 7 line 278).

·         Comment no. 4

R: This reviewer has many years of experience in using statistical methods of analysis and is very dubious that "a sample size of at least 10 subjects in each group would yield at least 90% power of detecting the significant difference."

AU: Thank you for your comment. As we mentioned in our manuscript, the sample size was determined according to vaspin concentration based on the article described by Teshigawara et al. (DOI: 10.1210/jc.2011-3297.). The calculations showed, that a sample size of at least 10 subjects in each group would yield at least 90% power of detecting the significant difference. We verified this calculations one more time (page 4 line 153).

R_new: This reviewer is not acquainted with the Teshigawara paper. However, the referenced page in the revised manuscript does not provide any explanations how this sample size was estimated. The authors in such a case required to provide information on the expected sample specific means and SDs

·         Comment no. 5

R:" The study sample includes individuals with a wide range of ages (in the obese group mean age was 43.3 and SD -13.4). In view of this reviewer, in all the comparisons and correlations, the effect of age must be taken into account. Due to the small sample size, probably in advance adjustment for best fitting age function of each independent variable in the study would be recommended.

AU: The correction has been added to the Table 2 (page 5 line 215).

R_new: This reviewer found no consideration of the adjusted for age (age and BMI) correlations in the revised Results section.

·         Comment no. 6

R: It should be mentioned that various body/fat mass measurements are expected to be highly inter-correlated (in multivariate regression analysis, they will create heavy redundancy due to collinearity) and, therefore, shouldn't be all analyzed. Probably most independent variables should be selected and examined.

AU: Thank you for your comment. We are aware that a multivariate regression may cause a heavy redundancy due to collinearity, however we use a logistic regression with one variable (vaspin concentration only).

·         Comment no. 7

R: The results of the analyses presented in Table 2 are problematic in many instances, as may be explained by their covariation with age and BMI. For example, there are many studies suggesting that hsCRP and IL6 levels correlate with BMI and other obesity indices. This would explain their and other variables' correlations with vaspin, in particular when obese and lean individuals combined. Of course, body weight and BMI, as well as waist and hip circumferences, should not be both tested, etc. WHR is probably better in any case.

Therefore, in view of this reviewer, Table 2 results presentation will benefit if two columns were added: the same type of analysis after adjustment for age and the same type of analysis after adjustment for age and BMI.

AU: Table 2 has been modified as suggested. WHR has been introduced instead of waist and hip circumferences (page 5 line 215).

R_new: The same as above. This reviewer found no consideration of the adjusted for age (age and BMI) correlations in the revised Results section.

·         Comment no. 8

R: There are numerous tests of the same hypothesis in the study (multiple testing problem). Therefore, correction for multiple testing is required.

AU: Thank you for your suggestions. Although, due to the comparative analysis between the two groups, that is the study and the control, we decided not to introduce the corrections to the testing. In the future studies, with more groups, we certainly use the correction for multiple testing.

R_New: The authors tested the same working (null hypothesis: no differences between the samples) several times. Therefore correction for multiple testing is required.

Round 3

Reviewer 2 Report

This reviewer believes that the authors of the study made a serious effort to meet his suggestions and recommendations and significantly improved the data presentation and the manuscript in general. I am still concerned with a small sample size and therefore recommend stressing in the discussion section and in the abstract of the paper that due to the small sample size, the provided results should be considered preliminary and further studies with a bigger sample size are needed to confirm the reported results.

The minor but important point concerns Table 2. Three columns of correlations between vaspin and covariates are provided, but no explanations are given in the legends of the table. Should be provided. 

It should also be mentioned that due to significant intercorrelations between the obesity-related variables, it is not surprising that their adjustment for BMI diminishes drastically their independent correlations with the vaspin levels.
